# FLUTE: A Scalable, Extensible Framework for High-Performance Federated Learning Simulations

**Dimitrios Dimitriadis**
Microsoft Research
didimit@microsoft.com

**Mirian Hipolito Garcia**
Microsoft Research
mirianh@microsoft.com

**Daniel Madrigal Diaz**
Microsoft Research
danielmad@microsoft.com

**Andre Manoel**
Microsoft Research
amonteiroman@microsoft.com

**Robert Sim**
Microsoft Research
rsim@microsoft.com

## Abstract

In this paper we introduce "Federated Learning Utilities and Tools for Experimentation" (FLUTE), a high-performance open source platform for federated learning research and offline simulations. The goal of FLUTE is to enable rapid prototyping and simulation of new federated learning algorithms at scale, including novel optimization, privacy, and communications strategies. We describe the architecture of FLUTE, enabling arbitrary federated modeling schemes to be realized, we compare the platform with other state-of-the-art frameworks, and we describe available features of FLUTE for experimentation in core areas of active research, such as optimization, privacy, and scalability. A comparison with other established platforms shows speed-ups up to $42\times$ and savings in memory footprint of $3\times$. A sample of the platform capabilities is presented for a range of tasks and other functionality such as scaling and a variety of federated optimizers.

## 1 Introduction

*Federated Learning* (FL) has been proposed as a strategy to address new constraints in data management, driven by the need for privacy compliance for personal data and information, as well as widely distributed and segregated data silos McMahan et al. [2017]. FL is a decentralized machine learning scheme with a focus on collaborative training and user data privacy. The key idea is to enable training of a global model with the participation of multiple clients coordinated without the need of sharing the client local information. In contrast to other forms of Distributed Training (DT) Ben-Nun and Hoefler [2019], Sergeev and Bals [2018], Abadi et al. [2016a], Chen and Huo [2016], the clients train a version of the model on segregated local data. Then, the tuned parameters are sent back to the server where the global model is updated by aggregating these local client information.

One of the challenges when building Federated Learning platforms is the need for scaling the learning process to millions of clients, in order to simulate real-world conditions under reasonable computing resources. As such, developing and validating any novel algorithm in realistic circumstances, e.g., using real devices or close-to-real scaled deployments, can be particularly difficult. Simulation platforms play an important role, enabling researchers and developers to develop proof-of-concept implementations (POCs) and validate their performance before building and deploying in production. While several open-source frameworks have been developed to enable FL solutions, few optimize performance for scientific agility and offline scalability.

Workshop on Federated Learning: Recent Advances and New Challenges, in Conjunction with NeurIPS 2022 (FL-NeurIPS'22). This workshop does not have official proceedings and this paper is non-archival.

This paper introduces a novel platform "*Federated Learning Utilities and Tools for Experimentation*" (FLUTE) as a framework for running large-scale, offline FL simulations. It is designed to be flexible, to enable arbitrary model architectures[1], and to allow for prototyping novel approaches in federation, optimization, quantization, privacy, and so on. Finally, it provides an optional integration with AzureML workspaces AzureML Team [2016], enabling scenarios closer to real-world applications, and leveraging platform features to manage and track experiments, parameter sweeps, and model snapshots.

The main contributions of FLUTE are:

1. A platform for high-performance FL simulations at scale (scaling to millions of clients),

2. Flexibility in the platform to include new FL paradigms, unlocking research, experimentation, and proof-of-concept (PoC) development,

3. A generic API for new model types and data formats,

4. A pre-built list of features - state-of-the-art federation algorithms, optimizers, differential privacy, bandwidth management, client management/sampling, etc,

5. Experimental results illustrating the utility of the platform for FL research,

6. A comprehensive analysis and comparisons with two of the leading FL simulation platforms[2], i.e. FedML He et al. [2020] and Flower Beutel et al. [2020].

The goal of FLUTE is to facilitate the study of new algorithmic paradigms and optimizations, enabling more effective FL solutions in real-world deployments, by extension the platform has been applied by multiple research groups to advance their ideas, such as: Dimitriadis et al. [2020a], Liu et al. [2022a], Sun et al. [2022] and Wang et al. [2022]. In contrast to incumbent platforms, we decouple FLUTE from production constraints such as process isolation or data security measures. **This decoupling is a feature, rather than a bug:** by focusing on scaling and performance we can optimize for scientific agility, enabling researchers to experiment and prototype much more efficiently. The unique architecture of FLUTE allows clients to be instantiated on-the-fly once the resource is available and then process each independent client asynchronously, making it more efficient than other platforms. On the other hand, FLUTE does not currently address challenges like data collection, secure aggregation or attestation. The code for the platform is open-sourced and available at **aka.ms/msrflute**.

## 2   Background and Prior Work

In general, there are two different approaches concerning the architecture of FL systems: either using a central server [Patarasuk and Yuan, 2009], as the "coordinator or orchestrator", or opting for peer-to-peer learning, without the need of a central server [Liang et al., 2020]. FLUTE is based on the "server-client" architecture, where the server coordinates any number of clients. Besides the basic architecture, FLUTE addresses technical challenges such as the required resources, i.e. bandwidth, and computing power, efficiency, optimization and learning pipeline [Shamir et al., 2013], and privacy constraints. Such challenges are usually attributed to either the ML side of federated learning, e.g., the distributed nature of the tasks, or the engineering side where the available resources are limited.

**Communication overhead:**   FL relies heavily on the communication between server and the clients to complete any training iteration. The fact that some of the clients and the server can be in different networks may cause limited connectivity, high latency, and other failures. Different approaches have been proposed, e.g., gradient quantization and sparsification [McMahan et al., 2017, Jhunjhunwala et al., 2021], different architectures per client [Cho et al., 2022], use of adapters for federating transformer models, etc. Most of these approaches are already implemented in FLUTE, e.g., quantization results shown in Section 6.

---

[1]The repository provides some examples and users are urged to add their implementations.

[2]Based on the availability of simulation functionality and the number of downloads from their GitHub repository.

**Hardware heterogeneity:** Computing capabilities of each client can vary, i.e., CPU, memory, battery level, storage are not expected to be the same across all nodes. This can affect both the selection and availability of the participating devices and it can bias the learning process. Different approaches have been also proposed to address slower clients, i.e. "stragglers", with the most popular allowing for asynchronous updates and client dropouts. FLUTE provides a flexible, asynchronous framework to incorporate workers of different capabilities. Also, there is an intuitive way of modeling faster/slower nodes as part of the training process.

**Unbalanced and/or non-IID data:** Local training data are individually generated according to the client usage, e.g., users spending more time on their devices tend to generate more training data than others. Therefore, it is expected that these locally segregated training sets may not be either a representative sample of the global data distribution or uniformly distributed between clients. A simple strategy to overcome communication overheads was proposed with the "*Federated Averaging*" (FedAvg) algorithm [McMahan et al., 2017]. In this approach, the clients perform several training iterations, and then send the updated models back to the server for aggregation based on a weighted average. FedAvg is one of the go-to training strategies for FL, given the simplicity and the consistently good results achieved in multiple experiments. On the other hand, FedAvg is not the best aggregation strategy, especially in the case of non-IID local data distributions. Over time, new approaches have emerged to overcome these limitations, for example, adaptive optimizers [Reddi et al., 2021], SCAFFOLD [Karimireddy et al., 2020], and the Dynamic Gradient Aggregation (DGA) algorithm [Dimitriadis et al., 2020b], which propose optimization strategies to address the heterogeneity problems on data and devices.

**Threats:** The ever-increasing interest for applying FL in different scenarios has brought interest in malicious attacks and threat models, as described [Liu et al., 2022b]. FL itself cannot assure either data privacy, or robustness to diverse attacks proven to be effective in breaking privacy or destroying the learning process. Without any mitigation, both the server and clients can be attacked by malicious users, e.g., attackers can poison the model by sending back to the server fake model parameters [Zhang et al., 2021] or fake the server and send a malicious model to the clients, stealing local information [Enthoven and Al-Ars, 2020]. As such, FL strategies started to incorporate techniques like Differential Privacy (DP), as detailed in [Wei et al., 2020] or Multi-Party Computation (MPC), which only reveals the computation result while maintaining the confidentiality of all the intermediate processes [Byrd and Polychroniadou, 2020, Bhowmick et al., 2019]. Some defenses against inference and backdoor attacks have been implemented in FLUTE Wang et al. [2022].

**Simulation and prototyping:** Building federated learning solutions can require significant up-front engineering investment, often with an unclear or uncertain outcome. Simulation frameworks enable FL researchers and engineers to estimate the potential utility of a particular solution, and investigate novel approaches, before making any significant investments. To this day, different FL platforms have been proposed, however, most of them have been designed with a specific purpose which limits experimentation agility for complex large-scale FL scenarios. Some frameworks have been developed aiming to support researchers such as: *TensorFlow Federated* [Abadi et al., 2016a] and *PySyft* [Ziller et al., 2021] whose main focus is privacy, also some simulators have been introduced within a production environment for proof-of-concept scenarios including *FedML* [He et al., 2020] and *Flower* [Beutel et al., 2020]. Finally we have some frameworks that are introduced to provide a wide benchmark of datasets and metrics like *LEAF* [Caldas et al., 2018]. An extensive comparison of the most popular frameworks is shown in Section 5, evaluating the competitive advantages of FLUTE vs some of the most representative platforms based on the number of stars on GitHub: *FedML* and *Flower*.

## 3 FLUTE Platform Design

FLUTE is designed as a scalable framework for rapid prototyping, encouraging researchers to propose novel FL solutions for real-world applications, in scale, data volume, etc, under the following design constraints/specs:

- *Scalability*: Capacity to processs many thousands of clients on any given round. FLUTE allows to run large-scale experiments using any number of clients with reasonable turn-

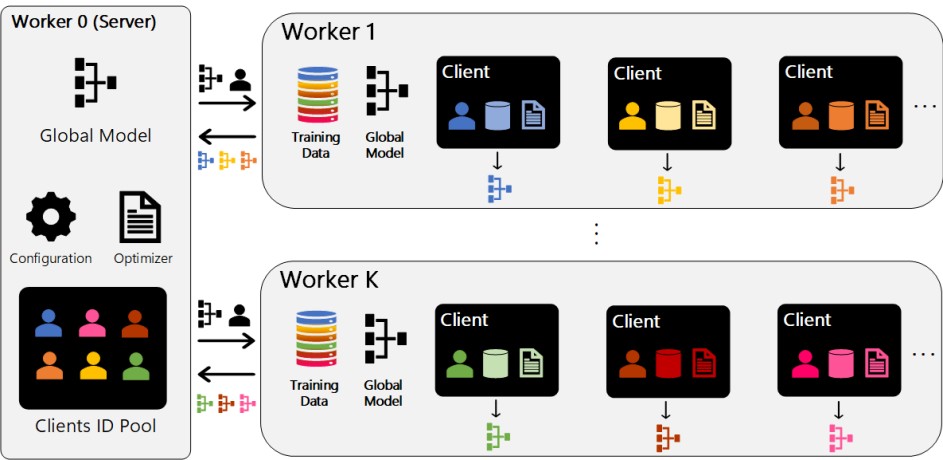

Figure 1: Client-server communication protocol. At each iteration, the server sends a copy of the global model to each worker, samples a number of clients and asynchronously assigns them to a worker as they become available

around time, since scale is a critical factor in understanding practical metrics such as convergence and privacy-utility trade-offs.

- *Flexibility*: Allow for any combination of model, dataset and optimizer. FLUTE supports diverse FL configurations, including some of the state-of-the-art algorithms such as DGA Dimitriadis et al. [2020b], FedAdam Reddi et al. [2020] and FedAvg McMahan et al. [2017], with Pytorch being the framework of choice for implementing the training pipelines.

- *Expandable*: Enable end users to incorporate new model architectures and scenarios, and allow them to easily plug in customized/novel techniques like differential privacy, gradient quantization, transfer learning, and personalization. FLUTE provides an open architecture allowing the incorporation of new algorithms and models in a straightforward fashion.

FLUTE design is based on a central server architecture, as depicted in Figure 1. The logical workflow is:

1. Send an initial global model to participating clients,
2. Train instances of the global model with the locally available data on each client,
3. Send training information, e.g., updated model parameters, logits (if required), and/or gradients/pseudo-gradients back to the server,
4. Combine the returned information on the server to produce a new global model,
5. Optionally, update the global model with an additional server-side rehearsal step,
6. Send the updated global model to the next sampled subset of clients,
7. Repeat steps 2 - 6 for the next training iteration.

A FLUTE job leverages one or more multi-GPU computers (local or in the cloud) running up to a total of $K$ worker processes, each executing tasks assigned by the server (Worker 0). The number of workers is decoupled from the number of clients, $P$, allowing the platform to scale to millions of clients even when $K \ll P$. During training, the server plays the role of both the *orchestrator* and the *aggregator*. First, it distributes a copy of the global model(s), training data and the list of all the client-IDs among the workers. The workers iteratively process the clients' data producing new models, and send the models back to the server. After $N$ of the clients are processed, the server aggregates the resulting models, typically into a single global model. Algorithm 1 describes in detail this process and Figure 2 exemplifies the execution flow.

The distributed nature of FLUTE is based on PyTorch, using the `torch.distributed` package as the communication backbone. The interface between server and workers is based on message payloads containing model parameters, client-IDs and training instructions. There are four message-types that can be passed from server to worker:

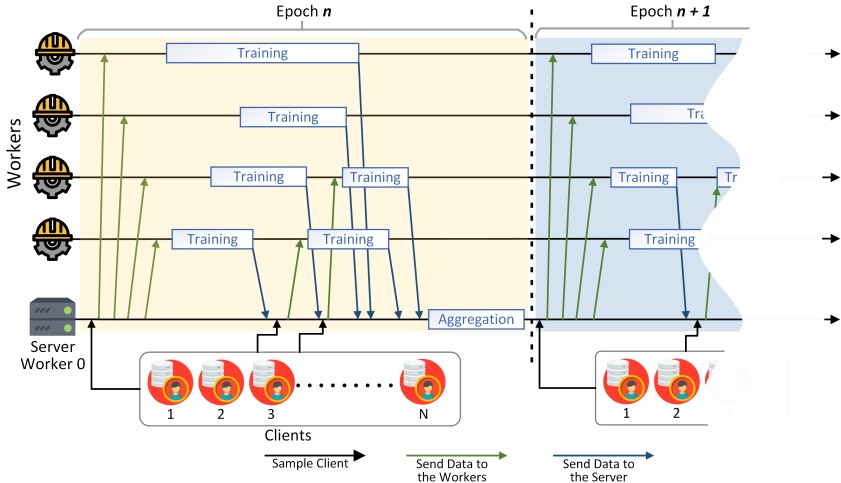

Figure 2: FLUTE Execution Flow: The server samples $\mathcal{N}$ of the clients and sends them to the K workers. Every time one of the workers finishes processing the client data, it returns the gradient and draws the next client until all clients are processed.

---

**Algorithm 1** FLUTE Orchestration: $\mathcal{P}$ is a Client Pool, which contains IDs of each client, $P = |\mathcal{P}|$, $M$ is the federation rounds to be executed, $K$ is the number of Workers, $\mathcal{N}$ is the subset of clients per iteration and $N = |\mathcal{N}|$ the number of clients per iteration

---

**Server-Side Worker-0:**
**for** each federated round from $1, \ldots, M$ **do**
  $\mathcal{N} \leftarrow$ Sample $N$ clients from $\mathcal{P}$
  **repeat**
    Wait for $worker_k$ to finish
    Save pseudo-gradient response from $worker_k$
    $c \leftarrow$ Sample client-ID from $\mathcal{C}$
    Dispatch model and $c$ to $worker_k$
  **until** all client-IDs $c$ in $\mathcal{N}$ have been processed
  Aggregate pseudo-gradients
  Update model with optimization step
**end for**

**Client-Side Worker-k**, with k > 0:
Load client and model data
Execute local training processes
Send pseudo-gradients and statistics about local training to Worker-0

---

1. `Update` creates a copy of the model parameters and learning rate on the worker,

2. `Train` triggers the execution of a training step on a worker, for a given client. The resulting model (or pseudo-gradient) is passed from worker to server,

3. `Evaluate` triggers the execution of an evaluation step in a given client. Resulting metrics are passed from worker to server,

4. `Terminate` shuts down the worker thread.

FLUTE leverages the communication scheme in Figure 1 by loading a local copy of the training data on each worker prior to training, significantly reducing the traffic communication between server and workers, only sending client indices. As previously noted, while this design decision would not be viable in production, it serves as an important performance optimization that has no impact on the validity of simulation results. Clients are implemented as isolated object instances– local data for each client stays within its logical boundaries and it is never sent to the server or aggregated with

other local data sources. Only metrics, model parameters or gradients are communicated between the server and clients – these quantities can be encrypted if necessary [3].

# 4 FLUTE Features

FLUTE provides a range of built-in functionality while state-of-the-art algorithm implementations cover important areas in FL research. In this section we discuss federated optimization, differential privacy, bandwidth efficiency, personalization and computing resource capabilities of FLUTE.

## 4.1 Federated Optimization

The "Federated averaging" (FedAvg) algorithm, Konecny et al. [2015], McMahan et al. [2017] is the first and perhaps the most widely used FL training algorithm. The server samples $\mathcal{M}_T \subset \mathcal{N}$ of the available $\mathcal{N}$ devices and sends the model $\mathbf{w}_T^{(s)}$ at that current iteration $T$. Each client $j$, $j \in \mathcal{M}_T$ has a version of the model $\mathbf{w}_T^{(j)}$, where it is locally updated with the segregated local data. The size of the available data $\mathcal{D}_T^{(j)}$, per iteration $T$ and client $j$, is expected to differ and as such, $N_T^{(j)} = |\mathcal{D}_T^{(j)}|$ is the size of processed local training samples. After running $\mathcal{E}$ steps of SGD, the updated model $\hat{\mathbf{w}}_T^{(j)}$ is sent back to the server. The new model $\mathbf{w}_{T+1}$ is given by

$$\mathbf{w}_{T+1} \leftarrow \frac{1}{\sum_{j \in \mathcal{M}_T} N_T^{(j)}} \sum_{j \in \mathcal{M}_T} N_T^{(j)} \hat{\mathbf{w}}_T^{(j)} \tag{1}$$

The server-side model in iteration $T + 1$ is a weighted average of the locally updated models of the previous iteration. Despite some drawbacks like lack of fine-tuning or annealing of a global learning rate, FedAvg is the baseline aggregation approach in FLUTE, based on its popularity.

Although the golden standard in FL training, FedAvg presents several drawbacks Zhao et al. [2018]. A different family of learning algorithms, "Adaptive Federated Optimizers" has been proposed to address them Reddi et al. [2020], Dimitriadis et al. [2020b], Li et al. [2019a], where the clients return pseudo-gradients instead of model parameters.

$$\mathbf{w}_{T+1} \leftarrow \mathbf{w}_{T+1} - f\left(\sum_{j \in \mathcal{M}_T} \alpha^{(j)} \left(\hat{\mathbf{w}}_T^{(j)} - \mathbf{w}_{T-1}^{(s)}\right)\right) \tag{2}$$

where $f(\cdot)$ is the optimization function, $\hat{\mathbf{w}}_T^{(j)} - \mathbf{w}_{T-1}^{(s)}$ the difference between the fine-tuned local model and the seed model of the same iteration. Finally, the weights $\alpha$ can be 1 in the case of FedAdam or take different values, e.g., in the case of DGA, as detailed in Dimitriadis et al. [2021].

The training process consists of two optimization steps: first, on the client-side using a stateless optimizer for local SGD steps, and then on the server-side with a "global" optimizer utilizing the aggregated gradient estimates. The two-level optimization provides both speed-ups in convergence rates due to the second optimizer on the server, and improved control over the training by adjusting the learning rates. In addition, scaling in the number of clients becomes straightforward by adjusting the server-side optimizer.

The FLUTE system provides support for this group of federated optimizers by adjusting the gradient aggregation weights, and the server-side optimizers, making FedYogi and FedAdam Reddi et al. [2020], and DGA Dimitriadis et al. [2021] rather straightforward to apply. Besides these optimizers, FLUTE provides implementations for FedProx, Li et al. [2019b] and SCAFFOLD, Karimireddy et al. [2020]. Finally, the FLUTE client scaling capabilities can be enhanced by switching to large-batch optimizers on the server-side, like LAMB You et al. [2020], and LARS You et al. [2017], as validated experimentally in Section 6.

---

[3]Encryption and secure aggregation are currently not implemented in FLUTE since these security mechanisms aren't necessary for simulating the learning process, and they are outside the scope of our research focus.

## 4.2 Differential Privacy

In the FL context, differential privacy (DP) Dwork et al. [2014] is typically enforced by clipping the norm of, and adding noise to, the gradients produced during training Abadi et al. [2016b]. This can be done either by each client (local DP), or by the server (global DP).

In FLUTE, either local or global DP can be used, depending on whether the clients or the server are responsible for doing the clipping and noise addition. In both cases, that is done directly to the pseudo-gradient, i.e., the difference between current and previous weights after each user's data is processed. The pseudo-gradients are re-scaled so that their norm is at most $C$, ensuring the norm of the difference between any two of them (the sensitivity) is bounded. We typically use Gaussian noise, with variance $\sigma^2 = 2\log\left(\frac{1.25}{\delta}\right)\frac{C^2}{\epsilon^2}$ picked so that the aggregation is at most $(\epsilon, \delta)$-DP w.r.t. each client. In the case of DGA Dimitriadis et al. [2020b], the aggregation weights are data-dependent, and also go through the same procedure.

Local clipping and noise addition can be accumulated as if it were per-example (i.e. per-client, or per-pseudo-gradient) clipping and noise addition in centra DP-stochastic gradient descent (DP-SGD). As such, we track the per-client noise across all clients and apply the Renyi DP accountant globally Mironov [2017]. Note that tighter bounds have been presented recently in the literature Gopi et al. [2021], Zhu et al. [2021] improving these results.

## 4.3 Bandwidth Efficiency

Gradients produced during training of the neural networks are known to be quite redundant, and can be compressed without adversely affecting the optimization procedure, e.g., gradient components can be represented by a single bit Seide et al. [2014], or some of the components discarded, making the gradient sparse Wangni et al. [2018]. Such compression leads to decreased bandwidth requirements.

In FLUTE, we use an approach similar to that of Alistarh et al. [2017]. We first get the dynamic range of the gradient components, and then create a histogram of $2^B$ bins between these two values, per layer. Next, we replace each gradient component by the label of the closest bin. That way, only the bin indices, together with the min./max. values need to be communicated. This quantization procedure is done on the client side, improving the uplink communication. We can also sparsify the gradients in parallel by keeping only the $p\%$ largest (in absolute value) components. If quantization is also active, binning is done before sparsification, but the original value of the component is used to decide whether it is replaced by the bin label or zeroed-out.

As part of FLUTE, novel compression algorithms are proposed, e.g., the use of adapters Houlsby et al. [2019] when the global models are based on transformers, achieving even further bandwidth compression, or the use of heterogeneous model architectures, as in Cho et al. [2022]. In more detail, the transformer models are communicated only once and frozen, while the adapter modules are federated. Despite the significant bandwidth benefits, the computation complexity is increased since all model parameters need to be evaluated. The results are presented in Section 6.

## 4.4 Personalization

The convergence of most Federated Learning optimization algorithms is theoretically proven when the client data distributions are iid. However, scenarios where the data distributions are non-iid, are far more challenging, e.g., Li et al. [2019b]. One of the different approaches for addressing this issue is with convex interpolation between the global $\boldsymbol{\theta}^{(r)}$ and the local models $\boldsymbol{\theta}_i^{(r,B)}$ for the $i^{th}$ client, Deng et al. [2020], after $B$ local training epochs. The resulting personalized model $\boldsymbol{\theta}_{int}^{(r)}$ after interpolation is given by

$$\boldsymbol{\theta}_{int}^{(r)} = \alpha_i \cdot \boldsymbol{\theta}_i^{(r,B)} + (1 - \alpha_i) \cdot \theta^{(r)} \tag{3}$$

and the interpolation weights $\alpha_i$ for each client $i$ are estimated as described in Deng et al. [2020]. FLUTE architecture supports this feature for new models, leveraging clients' local data to obtain models that are better adjusted to the local data distribution. A baseline experiment is included in the repository for computer vision (CV) tasks, as shown in Section 6.5. As expected, a certain drawback of this method is that it requires $P + 1$ models for federation with $P$ clients.

| | FLUTE | FedML (Parrot) | Flower (Simulator) |
|---|---|---|---|
| Focus | Research and Simulation | Research and Production | Research and Production |
| ML Framework | PyTorch | PyTorch | PyTorch / TensorFlow |
| Communication Protocols Supported | Gloo,NCCL | SP, MPI | Ray, gRPC |
| Support Security/Privacy related functions | ✓ | ✓ | ✓ |
| Support Multiple Federated Optimization Techniques | ✓ | ✓ | ✓ |
| Flexible and Generic API | ✓ | ✓ | ✓ |
| Cloud Integration | ✓ | ✓ | ✗ |
| Multi-GPU Support | ✓ | ✓ | ✗ |
| Performance Optimizations | ✓ | ✗ | ✗ |
| Easily Extensible to Production | ✗ | ✓ | ✓ |

Table 1: Comparison between FLUTE and Popular Federated Learning Simulation Platforms. This analysis is focused on the simulators provided by these platforms only.

## 4.5 Computing Resources

AzureML (AML) AzureML Team [2016] is the preferred computing FLUTE environment for staging, executing, tracking, and summarizing FL experiments. FLUTE has native AML integration for job submissions, allowing the users to use the built-in CLI or web interface for job/experiment tracking, and for visualization purposes. While FLUTE needs only the experiment-related configurations, AML expects the computing environment parameters such as target, cluster, code, etc. Besides AML, FLUTE can also run seamlessly on stand-alone devices, such as laptop and desktop machines (like those used for Section 5), using the local GPUs if/when available.

## 5 Comparison with related platforms

Multiple FL platforms have been proposed. However, most of them have been designed with a specific purpose limiting their flexibility to experiment with restricted resources in an acceptable turnaround time. Some frameworks oriented towards implementation allow researchers to work in a simulation environment using a production-compatible platform. Nonetheless, these simulators suffer from long runtimes, especially in complex FL scenarios, given that their architecture requires some production-oriented routines. Table 1 shows a detailed comparison of some of the most common [4] FL platforms' features and main focus.

FLUTE allows customized training procedures and complex algorithmic implementations at scale, making it a valuable tool to rapidly validate the feasibility of novel FL solutions, while avoiding the need to deal with complications that production environments present. FLUTE harnesses its architecture to provide a significant advantage in runtime and memory utilization, leveraging benefits of using NCCL Nvidia with GPUs, when available. For this comparison, we selected the *FedML* He et al. [2020] and *Flower* Beutel et al. [2020] platforms as the most representative, based on their number of stars on GitHub. In order to ensure a fair comparison among platforms, we used the datasets[5] and models[6] from the FedML Benchmarking Recipes[7] with the training configuration described in Table 2. All FLUTE scripts can be found under *Experiments* in the FLUTE repository.

| Model | Dataset | Algorithm | # Clients | Clients/round | Batch Size | Client Optim. | lr | Local Epochs | # Rounds | Test Freq |
|---|---|---|---|---|---|---|---|---|---|---|
| Log. Regr. | mnist | FedAvg | 1000 | 10 | 10 | SGD | 0.03 | 1 | 100 | 20 |
| CNN | fedmnist | FedAvg | 3400 | 10 | 20 | SGD | 0.1 | 1 | 800 | 50 |
| ResNet18 | fedcifar100 | FedAvg | 500 | 10 | 20 | SGD | 0.1 | 1 | 4000 | 50 |
| RNN | fedshakespeare | FedAvg | 715 | 10 | 4 | SGD | 0.8 | 1 | 1200 | 50 |

Table 2: Training configuration for FLUTE/FedML/Flower Benchmarking

Table 3 shows that FLUTE outperforms FedML by $42\times$ in speed and $3\times$ in memory consumption. The advantage of FLUTE relies on its ability to asynchronously assign new clients to the workers as

---

[4] Based on the number of stars on GitHub as of 10/26/22

[5] FedML Datasets https://github.com/FedML-AI/FedML/tree/master/python/fedml/data

[6] FedML Models https://github.com/FedML-AI/FedML/tree/master/python/examples/simulation/mpi_fedavg_datasets_and_models_example

[7] FedML Benchmarking Recipes and Results https://doc.fedml.ai/simulation/benchmark/BENCHMARK_MPI.html

they become available, and receive their outputs. On the other hand, FedML[8] links the number of workers with the number of MPI processes, which is reflected as the number of parallel clients during training, FLUTE design allows processing multiple clients per worker, decoupling the need for $1:1$ mapping between clients and training processes. In FLUTE, each worker holds a pre-loaded local copy of the training data, avoiding communication overheads during training as the Server only sends indices of the clients to instantiate.

|  | Task | | | | FedML (MPI) 0.7.303 | | | FLUTE (NCCL) 1.0.0 | | |
|---|---|---|---|---|---|---|---|---|---|---|
| Model | Dataset | Clients | Rounds | Acc | Time | GPU memory | Acc | Time | GPU memory |
| Log. Regr. | mnist | 1000 | 100 | 81 | 00:03:09 | 3060 MB | 81 | **00:01:35** | **1060 MB** |
| CNN | fedmnist | 3400 | 800 | 83 | 05:49:52 | 5180 MB | 83 | **00:08:22** | **1770 MB** |
| ResNet18 | fedcifar100 | 500 | 4000 | 34 | 15:55:36 | 5530 MB | 33 | **01:42:01** | **1900 MB** |
| RNN | fedshakespeare | 715 | 1200 | 57 | 06:46:21 | 3690 MB | 57 | **00:21:50** | **1270 MB** |

Table 3: GPU Performance comparison FLUTE 1.0.0 vs FedML 0.7.303 on 4x NVIDIA RTX A6000 using FedML Datasets. Test accuracy is reported from the last communication round.

An additional comparison of FLUTE versus Flower 1.0.0 [9] is presented in Table 4. While in principle Flower supports multi-GPU setups using Ray, we were unable to find any Flower/Ray configuration that out-performed the CPU setup, following the recipe provided on their GitHub for the simulator [10]. Thus, to run a fair comparison, we compare FLUTE CPU performance (using the Gloo backend) against Flower, evaluating the overall time of the job using the same setup for the *"MNIST Log. Reg."* task described in Table 2. FLUTE is up to $54\times$ faster than Flower on GPUs given that their simulation capabilities are not optimized for multi-GPU jobs out of the box.

Regardless, FLUTE, running on a Gloo backend, is $9\times$ faster than Flower, running only on CPUs.

# 6   Case Studies

This section provides some insights of the FLUTE features. The list of datasets, tasks and experimental results herein presented is by no means exhaustive. Also, no particular models are detailed since the platform allows training on any architecture currently supported by PyTorch. However, FLUTE provides the baseline configuration for the models that the platform supports and a group of datasets for the different tasks incorporated in the repository, inside the *Experiments* folder.

## 6.1   Sample of Baseline Experiments and Datasets

We provide some of the available models/tasks as part of the FLUTE distribution. This list of models and tasks is **not** exhaustive since the flexibility of the platform allows extensions in models such as Graph Neural Networks (GNNs), He et al. [2021], Gradient Boosted Trees (GBTs), Li et al. [2019c], and others:

- **ASR Task: LibriSpeech**: FLUTE offers a Speech Recognition template task based on the LibriSpeech task Panayotov et al. [2015]. The dataset contains about 1,000 hours of speech from 2,500 speakers reading books. Each of the speakers is labeled as a different client. In one of the ASR task examples, a sequence-to-sequence model was used for training, more details can be found in Dimitriadis et al. [2020a].

- **Computer Vision Task: MNIST and EMNIST**: Two different datasets, i.e., the MNIST LeCun and Cortes [2010] and the EMNIST Cohen et al. [2017] dataset are used for Computer Vision tasks. The EMNIST dataset is a set of handwritten characters and digits captured and converted to $28 \times 28$ pixel images maintaining the image format and data-structure and directly matching the MNIST dataset. Among the many splits of EMNIST dataset, we use the "EMNIST Balanced", containing ~132k images with 47 balanced classes.

- **NLP Tasks: Reddit**: Various NLP tasks are supported in FLUTE, e.g., 2 use-cases for MLM and next-word prediction using Reddit data Baumgartner et al. [2020]. The Reddit

---

[8]FedML Simulator (Parrot) on its release 0.7.303, commit ID 8f7f261f

[9]Flower Simulator on its release 1.0.0, commit ID 4e7fad9

[10]GitHub issue: PyTorch Simulation example with GPUs https://github.com/adap/flower/issues/1415

|  | FLUTE 1.0.0 Gloo/NCCL | | Flower 1.0.0 Ray | |
| --- | --- | --- | --- | --- |
| Accelerator | Acc | Time | Acc | Time |
| CPU | 80 | **00:03:20** | 80 | 00:30:14 |
| GPU 2x | 80 | **00:01:31** | 80 | 01:21:44 |
| GPU 4x | 81 | **00:01:26** | 79 | 00:56:45 |

Table 4: Performance comparison FLUTE 1.0.0 vs Flower 1.0.0 on 4x NVIDIA RTX A6000, AMD EPYC 7V12 64-Core Processor. Test accuracy is reported from the last communication round.

dataset consists of users' tweets grouped in months as published. For these use-cases, we use 2 months of Reddit data with 2.2M users. The seed models used are either from HuggingFace or a baseline language model (LM), as described below.

- **Sentiment Analysis: sent140, IMDb, YELP**: Sent140 Go et al. [2009], is a sentiment analysis dataset consisting of tweets, automatically annotated from the emojis found in them. The dataset consists of 255k users, with mean length of 3.5 samples per user. IMDb is based on movie reviews of 1012 users providing 140k reviews with 10 rating classes Diao et al. [2014]. The YELP dataset is based on restaurant reviews with labels from 1 to 5 Tang et al. [2015]. It contains 2.5k users with 425k reviews.

- **Baseline LM Model**: A baseline LM model is used for most of the experiments in Section 6. A two-layer GRU with 512 hidden units, 10,000 word vocabulary, and embedding dimension 160 is used for fine-tuning during the FL experiments. The seed model is pretrained on the Google News corpus Gu et al. [2020].

## 6.2    Quantization Experiment

The accuracy for a next-word-prediction task on the Reddit dataset and the baseline LM model (as described in Section 6.1) for various levels of quantization $B$ is shown in Table 5. As expected, using less bits leads to decreased performance in terms of accuracy.

|  | Quant. (bits) | Acc @1 (%) | Rel. Imprv. (%) |
| --- | --- | --- | --- |
| Seed Model | N/A | 9.83 | (56.62) |
| Server-Side Training | N/A | 22.30 | (1.59) |
| FL Train. | 32 | 22.70 | 0 |
|  | 10 | 22.40 | (1.32) |
|  | 8 | 22.20 | (2.25) |
|  | 4 | 21.30 | (5.87) |
|  | 3 | 18.80 | (17.21) |
|  | 2 | 17.80 | (21.58) |

Table 5: Next-word prediction: Top-1 accuracy after gradient quantization. The number of bits per gradient coefficient varies $2 - 32$.

We have also done experiments varying the sparsity level, while keeping the quantization constant at 8 bits, cf. Table 6.2. In this particular experiment, we had gains in bandwidth of up to 16x with no significant change in performance. Error compensation techniques Strom [2015] could be attempted in order to increase the performance at higher sparsity levels. The difference in performances for the case of 8-bit quantization level in Tables 5 and 6.2 is due to noise during the training process.

## 6.3    Performance for Variable/Different Number of Clients

The number of clients processed at each round is a variable we can control on FLUTE. Here, we show in Table 7 a simulation with 1 server + 3 workers attached to RTX A6000 GPUs and 2.45GHz AMD EPYC cores for varying number of clients per iteration. Since clients are processed sequentially by each worker, runtime scales linearly. FLUTE also provides options for further speed-ups by processing clients in multiple threads and pre-encoding the data.

| % Sparsity | Gain in Bandwidth | Acc @1 (%) |
|---|---|---|
| 0.0 | 4x | 22.60 |
| 75.0 | 16x | 21.70 |
| 95.0 | 80x | 19.00 |
| 99.0 | 400x | 17.70 |

Table 6: Next-word accuracy obtained by varying sparsity level on gradients while keeping quantization fixed at 8 bits – gains in bandwidth are relative to standard 32 bits gradient. The performance reported is the best one over 5000 iterations, with 1000 clients being processed at each iteration.

| Number of Clients | Runtime (sec.) |
|---|---|
| 1,000 | $22.1 \pm 0.6$ |
| 5,000 | $111.3 \pm 2.4$ |
| 10,000 | $219.0 \pm 2.3$ |
| 50,000 | $1103.7 \pm 11.3$ |

Table 7: How long it takes for 3 workers to process different number of clients, on an NLG experiment using a GRU model and the Reddit dataset. Averages are computed over 20 iterations.

Table 7 shows that FLUTE scales gracefully the number of clients per iteration, without any upper bound to that number. We can also look at the predictive performance attained for different numbers of clients, and study how it changes as a function of the optimizer used.

| Num. of Clients | Optimizer local-server | Acc @ 1 (%) |
|---|---|---|
| No Fine-tuning | Seed Model | 9.80 |
| 1k clients/iter | SGD-Adam (Baseline) | 22.70 |
| | SGD-RL-based DGA | 22.80 |
| 10k clients/iter | SGD-Adam (Baseline) | 20.80 |
| | SGD-LARS | 17.00 |
| | Adam-LARS | 21.40 |
| | SGD-LAMB | 23.00 |
| Variable number $[5k-10k]$ clients/iter | SGD-Adam | 22.30 |

Table 8: Next-word Prediction task: Top-1 accuracy achieved varying number of clients and optimizers.

In Table 8 , we compare 4 different scenarios for optimizers, increasing the number of clients, showing that the accuracy remains stable for most of them. However, the Adam optimizer decreases its accuracy as the number of clients increase, compared to SGD-LAMB that reaches a better performance with a larger number of clients.

## 6.4 Comparing Optimizers

This experiment of next-word prediction, using the Reddit dataset and baseline LM model described in Section 6.1, explores model training performance for a variety of state-of-the-art optimizer choices. We trained a recurrent language model, fixing the number of clients per round to 1,000, and varying the choice of optimizer in the central aggregator. Specifically, we applied standard SGD Rosenblatt [1958], ADAM Kingma and Ba [2017], LAMB You et al. [2020], and LARS You et al. [2017]. Table 9 illustrates the performance of each optimizer, including maximum validation accuracy, and convergence rate: the number of rounds to reach 95% of the max. accuracy. Note there is no hyper-parameter tuning of the optimizers for this experiment.

## 6.5 Personalization Experiments

The CIFAR-10 task is used for the personalization experiments, splitting the data across 100 clients, and sampling 10 clients per iteration. The client data are split according to the process described

| Optimizer | Acc @1 (%) | Convergence Round |
|---|---|---|
| LAMB | 23.10 | 115 |
| ADAM | 22.70 | 641 |
| SGD | 20.60 | 2172 |
| LARS | 17.40 | 414 |

Table 9: Next-word prediction task: Top-1 Accuracy and training rounds to 95% convergence for various central optimizer choices.

in He et al. [2020], with $\alpha \in [0.2, 1.0]$ for the Dirichlet label distribution (client distributions are more iid when the $\alpha$ values are larger). In addition to the label distributions, we investigate different feature distributions by applying locally different image transformations (per each client). For this experiment, we fix the test samples to match the local training data/label distributions, i.e. we split the test set to follow similar local label distributions as the training samples. The image transformations are unique per client for both the training and test samples, when applicable. Herein, we investigate 3 different training strategies, i.e. a global model trained with DGA, local models trained with SGD and the convex interpolation of these two, as described in Section 4.4. The relative performance improvement shown in Table 6.5 is between the global and the interpolated models. Convex interpolation, as described in Section 4.4, always benefits overall performance but the gains

|  | Global | Local | Interp. | Rel. Imprv. |
|---|---|---|---|---|
| iid ($\alpha = 1.0$) | 74.12 | 46.10 | 77.72 | **13.91** |
| non-iid ($\alpha = 0.5$) | 72.33 | 54.90 | 79.56 | **26.13** |
| non-iid ($\alpha = 0.2$) | 69.50 | 70.70 | 85.43 | **52.23** |
| iid ($\alpha = 1.0$) + Feat. Transf. | 51.34 | 46.48 | 62.78 | **23.51** |
| non-iid ($\alpha = 0.5$) + Feat. Transf. | 49.40 | 54.57 | 67.60 | **35.97** |
| non-iid ($\alpha = 0.2$) + Feat. Transf. | 47.55 | 70.90 | 77.45 | **57.01** |

Table 10: Personalization on CIFAR-10: Two sources of non-iidness, (i) Label distribution based on $\alpha \in [0.2, 1.0]$ and (ii) Different image transformations per client. All reported results are in image classification accuracy (%).

are more significant in the case of extreme non-iidness.

Increasing the local non-iidness helps the *local* model performance (and the interpolated combination between local and global models). The local datasets have more examples of the particular labels since the total number of local training samples remains constant. As such, the local models can generalize better, improving the overall performance. The feature-based non-iidness doesn't affect the local models, since these models are trained on matched transformations, their impact on the model quality is minimal. On the contrary, image transformations have great impact on the global models due to the increased data mismatch.

## 7 Discussion and Conclusions

In recent years, researchers have made significant efforts to address the challenges in Federated Learning (FL), especially when it comes to setting up FL-friendly environments – privacy guarantees, time-consuming processes, communication costs and beyond. Herein, we presented FLUTE, a versatile, open-architecture platform for high-performance federated learning simulation geared towards the research community to streamline the process of prototyping groundbreaking algorithms in Federated Learning. FLUTE's novel architecture provides scaling capabilities, several state-of-the-art federation approaches and related features such as differential privacy and personalization, with a flexible API that allows to easily incorporate extensions and the introduction of novel approaches. FLUTE is model and task-independent, and provides facilities for easy integration of new model architectures based on PyTorch.

The goal of FLUTE is to optimize the available resources, to enable rapid experimentation and prototyping of novel algorithms, facilitating the development of new FL research efforts in the most expeditious manner. We encourage the research community to explore new research using FLUTE and invite contributions to the public source repository.

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

## A  Appendix – Stale Gradient Analysis

The FLUTE platform offers flexibility when sampling the participating clients from the pool of candidates. A range in the number of clients can be given, fluctuating between the two ends. This can be seen as "*client dropouts*", where a number of clients can be randomly discarded. Since the default optimization pipeline is based on DGA, as in Section 4.1, the learning rate can be adjusted accordingly.

Similar to the dropout functionality, FLUTE offers an option of delaying the contributions of random clients rather than discarding the corresponding gradients. The system can introduce a 1-step "staleness" to the system by randomly delaying a subset of the clients by 1 iteration. The convergence analysis for the stale gradients scenario is held in Appendix A. As shown, the error introduced due to staleness is upper bounded. As such, there is theoretical guarantee that the model will finally converge. The theoretical conclusions are experimentally verified using FLUTE, shown in Figure 3.

A complementary approach to deal with the issue of straggling is to use asynchronous SGD. In asynchronous SGD, any learner can evaluate the gradient and update the central PS without waiting for the other learners. Asynchronous variants of existing SGD algorithms have also been proposed and implemented in systems, e.g., Agarwal and Duchi [2011], Dutta et al. [2018]. In general, analyzing the convergence of asynchronous SGD with the number of iterations is difficult in itself because of the randomness of gradient staleness.

Gradient descent is a way to iteratively minimize this objective function by updating the parameter $w$ based on the gradient of the model $\theta_\tau^{(s)}$ at every iteration $\tau$, as given by

$$\theta_{t+1}^{(j)} = \theta_t^{(j)} - \eta^{(j)} \nabla_\theta \mathcal{L}_\theta(\mathbf{x}_i^{(j)}) \tag{4}$$

for the $j$ client, over the local data mini-batches $\mathbf{x}_i^{(j)}$. As described in Section 4.1 the clients are estimating a pseudo-gradient $\tilde{\mathbf{g}}_{T_j+\tau}^{(j)}$ at the end of their training cycle,

$$\tilde{\mathbf{g}}_{T_j+\tau}^{(j)} = \theta_{T_j}^{(j)} - \theta_\tau^{(s)} \tag{5}$$

where $T_j$ is the time took for the client $j$ to estimate the final local model, and $\theta_\tau$ is the global/initial model communicated to the client at time $\tau$. As in Dimitriadis et al. [2020b], these pseudo-gradients are weighted and aggregated

$$\theta_{\tau+1}^{(s)} = \theta_\tau^{(s)} - \eta^{(s)} \sum_{j \in N} \mathbb{I}_{j,\tau} \alpha_j \tilde{\mathbf{g}}_{T_j+\tau}^{(n)} \tag{6}$$

where $N$ the number of clients per iteration $\tau$, and the samples of

$$\mathbb{I}_{j,\tau} = \begin{cases} 1 & \text{if } T_j + \tau \in W_{[\tau,\tau+1)} \\ 0 & \text{else} \end{cases}$$

and $\widetilde{\mathbb{I}}_{j,\tau} = 1 - \mathbb{I}_{j,\tau}$

There are different degrees of staleness and for this work, the stale gradients are considered to fall at most one iteration behind, i.e. some of the gradients $\tilde{g}_{T_j+\tau-1}^{(j)}$ are part of the aggregation step in Eq. 6 for the window $W_{[\tau,\tau+1)}$. In other words, Eq. 6 now becomes

$$\theta_{\tau+1}^{(s)} = \theta_\tau^{(s)} - \eta^{(s)} \left[ \sum_{j \in J} \alpha_j (\theta_{T_j}^{(j)} - \theta_\tau^{(s)}) + \sum_{i \in I} \alpha_i (\theta_{T_i}^{(i)} - \theta_{\tau-1}^{(s)}) \right] \tag{7}$$

where $J$, $I$ is the index of nodes without/with stale gradients and assuming that $J \cup I = N$, i.e, the union of clients with current and stale gradients cover the client space per iteration. Assuming that the final models $\theta_{T_j}$ per client, would reach a similar point regardless of the starting model $\theta_\tau^{(s)}$ (a realistic assumption in convex models).

$$\theta_{\tau+1}^{(s)} \approx \theta_\tau^{(s)} - \eta^{(s)} \left[ \sum_{n \in N} \alpha_n (\theta_{T_n}^{(n)} - \theta_\tau^{(s)}) + \sum_{i \in I} \alpha_i (\theta_\tau^{(s)} - \theta_{\tau-1}^{(s)}) \right] \tag{8}$$

Based on Eqs. 6, 8, the stale gradients of the $I$ nodes introduces an error term $E_\tau$ which depends only on the weights $\alpha_i$ and the difference with the previous model, i.e, the aggregated gradients of the previous time-step,

$$E_\tau = \eta^{(s)} \left( \theta_\tau^{(s)} - \theta_{\tau-1}^{(s)} \right) \sum_{i \in I} \alpha_i$$
$$= \eta^{(s)} \left( \theta_\tau^{(s)} - \theta_{\tau-1}^{(s)} \right) \sum_{i \in I} \widetilde{\mathbb{I}}_{i,\tau} \alpha_i$$

The expectation of the $L2$-norm of the error is,

$$\mathbb{E}\left[ \|E_\tau\|_2 \right] =$$
$$= \mathbb{E}\left[ \left\| \eta^{(s)} \left( \theta_\tau^{(s)} - \theta_{\tau-1}^{(s)} \right) \sum_{i \in I} \widetilde{\mathbb{I}}_{i,\tau} \alpha_i \right\|_2 \right] \tag{9}$$
$$\leq \eta^{(s)} \mathbb{E}\left[ \left\| \left( \theta_\tau^{(s)} - \theta_{\tau-1}^{(s)} \right) \right\|_2 \right]$$

since $\sum \widetilde{\mathbb{I}}_{i,\tau} \alpha_i \leq 1$. According to Eq. 9, the upper-bound of the error term due to the stale gradients is the norm of the model differences between updates weighted by the learning rate $\eta^{(s)}$. In other words, the expectation of the norm of the error due to stale gradients is bound by the model updates (in fixed points in time). If we call $\Delta_\tau$ the norm of the difference between sequential in time models,

$$\Delta_\tau = \left\| \theta_\tau^{(s)} - \theta_{\tau-1}^{(s)} \right\|_2 \tag{10}$$

becomes smaller since the models converge to an optimal point. As such, $\lim_{\tau \to \infty} \Delta_\tau = 0$ and from Eq. 9, the error due to stale gradients becomes $\lim_{\tau \to \infty} E_\tau = 0$.

The conclusion from Eqs. 9 and 10 is in accordance with the analysis in Lian et al. [2015], where it is shown that the convergence rate does not depend on the staleness ratio given sufficient number

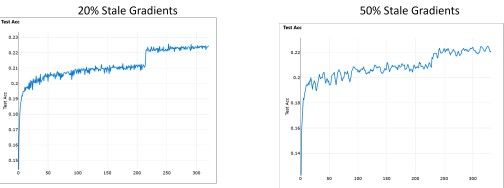

Figure 3: Next-word Prediction task: Top-1 Accuracy for Reddit dataset with Staleness or 1 iteration.

of iterations. It is proved that the benefits of not waiting for the strangler nodes (thus producing stale gradients) in terms of time needed to converge counter-balance the errors introduced early in the training process. Also, based on the analysis in Dutta et al. [2018], adjusting the learning rate schedule per iteration $\tau$ based on the staleness $\Delta_\tau$ can further expedite convergence,

$$\eta_\tau^{(s)} = \min\left\{\frac{C}{\Delta_\tau}, \ \eta_{\max}\right\}, \tag{11}$$

where $C$ is a predefined constant related to the error floor.

We experimentally verify the theoretical analysis in Appendix A with two different experiments, depending on the percentage of stale clients, 20% and 50% of the 1000 clients are stale – staleness in this experiment equals to 1 cycle. This experiment is based on the next-word prediction task using the Reddit dataset, together with the baseline LM model described in Section 6.1. As suggested, the model still converges to an optimal point in terms of accuracy. However, it takes longer for the case of 50% to reach a good point in performance.

