# OpenReview forum: "FLUTE: A Scalable, Extensible Framework for High-Performance Federated Learning Simulations"
_NeurIPS.cc/2022/Workshop/Federated_Learning — FL-NeurIPS 2022 Poster_

### Official Review · Reviewer_KWjL · 2022-10-17
**Review of Flute**

### Summary

The paper describes a new framework for simulating FL training in a "flexible" and "scalable" and "versatile" manner, FLUTE. FLUTE is based on PyTorch and supports multiple aggregators (FedAvg, DGA) and optimisers (SGD, ADam, LARS, LAMB), along with gradient quantisation, sparsificiation and differential privacy. On the system side, communication is based on Gloom/NCCL and provides integration with Azure cloud and multi-threaded support across GPUs. The authors report favourable performance compared to FedML (on GPU) and Flwr (on CPU), both in terms of end-to-end latency and memory consumption. Last, they provide integration with multiple datasets, performance of which they showcase in the appendix.

### Strengths

* The paper seems to be implementing an open-source, fully-fledged environment to facilitate FL research, with out-of-the-box support for many settings and features.
* The system seems to be easily extensible to support alternative settings, datasets and aggregators
* The experiments cover quite some breadth

### Weaknesses

* The novelty of FLUTE is minimal and the execution lacks refinement for production-level settings.
* The paper misses quite a lot of related work wrt federated frameworks and incorporated.
* The paper is overselling some features and evaluation is not fully documented or convincing at points.

### Detailed review

The paper reads like a high-level whitepaper. Although it reads easily, there are multiple missing details on the experimental setup and related work that make context-setting and comparisons weak. While performance seems better, the reasons behind these gains are hardly a contribution of the current system, and while a design decision, it is the same reason why FLUTE cannot run in production deployments.

### Further comments

* The authors claim from early on performance speed-ups of 42x and memory of 3x, which are mainly attributed to the decoupling of workers and clients (vs FedML) and Gloo communication backend (vs. Flower).
    * FedML, however, is able to run production-level workloads due to this modelling (worker = client).
    * I am not sure why the authors have selected to run Flower on CPU mode. Ray, the backend of simulated Flower, is able to share GPUs between different workers. At the same time, Flower also supports the (client = worker) paradigm, which FLUTE does not.
    * Appendix, "Flower [...] is fairly inefficient by design [...]": I would expect argumentation about this statement.
* Having a preloaded copy of the dataset goes along the same direction. Having the whole dataset available per client and loading on demand is more of a setup decision on each framework, rather than an innate characteristic of theirs. Simultaneously, this would not have been possible in a deployment setup for security and isolation purposes.

* Even though the authors have selected to compare with FedML and Flower as the most popular FL frameworks, they are missing references to other major frameworks, which could be given for context in background/related work.
    * Moreover, the authors should include a date for which the two frameworks were deemed as having the most "number of stars".

* "The FLUTE job leverages [...] VMs": Is there a requirement for running on virtual machines instead of bare metal?

* Table 1: multi-gpu support: I think flower has support for multi-gpu support through Ray.
* Table 2:
    * The accuracy for CIFAR-100 (federated) is quite low for 4000 rounds.
    * For Shakespeare, the performance metric selected is perplexity rather than accuracy.
    * I would have expected the comparison with Flower to be in the main text

* Appendix:
    * While the authors go the extra mile and describe extra datasets, they do not seem all to be used in the evaluation.
    * Are all "next-word prediction" tasks on the Reddit dataset? What are the hyperparameters used there?
    * Moreover, the authors assess the performance of extra components of FLUTE, but performance of most seems quite low for deployment standards (sub 25%) (and do not provide context for assessing relative to a baseline).

* Lacking information in experimental setup:
    * How many iterations have experiments been run for?
    * How are workers scheduled on the GPUs for runtime? How many workers are sharing the GPU at the same time?
    * Tables 2,4 are missing variance metrics.

### Phrasing

* "Encryption and secure aggregation [...] aren't strictly necessary for simulations": There could be research happening on this front, even in simulation settings. Current wording would preclude this from using FLUTE.
* Table 1, focus: Why isn't Flower's focus also simulation?
* Table 1, flexible and generic API: A bit a hand-wavy term
* Table 2 "lr" --> define abbreviation
* Figures 1,2 are quite small in size
* VMs --> "Virtual machines" (abbreviation not defined)
* The way execution asynchronicity is defined in text, it can be easily seen as asyncrhonous aggregation, which is not supported by FLUTE. I would urge the authors to refactor the respective description in the manuscript.
* I don't know if the NeurIPS checklist is needed for the workshop
* Tables usually have their caption above instead of below.
* Appendix, Table 5:
    * LDP, RDP undefined
    * Different number of decimals across rows. Maybe use alternative formatting
* Appendix, Quantisation experiment:
    * The authors are mixing two different concepts, i.e. quantisation, sparsity here. Maybe split since there is no interplay between the two?
    * Missing citations about the techniques
* l 464: validation accuracy --> test accuracy?
* Several abbreviations have not been defined throughout the text

---

### Official Review · Reviewer_kSGD · 2022-10-17
**FLUTE is a framework for scalable FL simulations**

FLUTE is a framework for FL research designed for high-performance simulations. The paper is well written and present the topics someone would expect from an new FL framework. However, I personally don't see a strong benefit in FLUTE compared to Flower, arguably its closest "competitor". Something that both seem to lack is a standardised set of datasets, configs and evaluation metrics (in essence a full training recipe) that would streamline the process of generating results and facilitate comparisons against alternative methods (e.g architecture variations, different aggregation functions, etc). Something along the [OGB effort](https://ogb.stanford.edu) but for FL would be incredibly valuable.

Weaknesses that need to be addressed::
*    The Authors state in Table 1 that Flower does not support multi-GPU training. This is not true, since it relies on Ray (mentioned in the table) in order to run multi-GPU and multi-node FL workloads. Fractional GPU allocation is supported by design and so is auto-scaling either on-premises clusters on in the cloud.
*    Comparisons to Flower are unfair (Table 4) in the sense that they were carried out not using Flower's simulation backend (which uses Ray). instead, these were done with the gRPC backend that is more suitable for real production deployments.

All in all, I'm not feeling very strong about this paper. From a framework paper I'd normally expect much more thorough comparison and clearly state (even if in a large table) what are the knobs offered by a framework (e.g. aggregation functions x,y,z; sampling criteria x,y,z; etc).

---

### Decision · Program_Chairs · 2022-10-20

Accept (Poster)